

# Modeling hadronization using machine learning

**Phil Ilten**[*], **Tony Menzo**[†], **Ahmed Youssef**[‡], **and Jure Zupan**[§]

Department of Physics, University of Cincinnati, Cincinnati, Ohio 45221, USA

[*] philten@cern.ch, [†] menzoad@mail.uc.edu, [‡] youssead@ucmail.uc.edu, [§] zupanje@ucmail.uc.edu

## Abstract

We present the first steps in the development of a new class of hadronization models utilizing machine learning techniques. We successfully implement, validate, and train a conditional sliced-Wasserstein autoencoder to replicate the PYTHIA generated kinematic distributions of first-hadron emissions, when the Lund string model of hadronization implemented in PYTHIA is restricted to the emissions of pions only. The trained models are then used to generate the full hadronization chains, with an IR cutoff energy imposed externally. The hadron multiplicities and cumulative kinematic distributions are shown to match the PYTHIA generated ones. We also discuss possible future generalizations of our results.

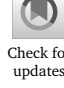

# 1 Introduction

A typical particle physics Monte Carlo event generator factorizes into three distinct steps or blocks of code: (i) the generation of the hard process, (ii) parton shower, and (iii) hadronization (including color reconnections). The first two steps are perturbative in their nature, and thus under good theoretical control, with significant efforts devoted to improving the precision even further [1–4]. The algorithmic challenges are efficient sampling of final state particle configurations, and taming the factorial growth of the calculations with the increasing number of particles. The simulation of the hard matrix element is performed either by a specialized code, e.g., MADGRAPH [5], which only calculates the hard process, or is directly included in complete event generators, such as PYTHIA [6], HERWIG [7], or SHERPA [8], that also perform the parton showering.

In contradistinction, the hadronization step is inherently non-perturbative. One is therefore forced to resort to phenomenological models inspired by non-perturbative descriptions such as lattice QCD. The two main models used in simulating hadronization are the Lund string model [9–11] and cluster model [12–14]. In the string model, quark–anti-quark pairs are thought of as being connected by a string, a flux tube of the strong force confined in the lateral direction. As the quark–anti-quark pair moves apart, the string breaks, creating new quark–anti-quark pairs in the process, resulting in the emission of hadrons. These emissions are performed iteratively, breaking the string either from the left or the right side, with the final step modified *post hoc* in order to provide an emission similar to the previous steps. This model requires extra parameters to describe the hadrons' transverse momenta and heavy particle suppression, and has some challenges describing baryon production. Over $\mathcal{O}(20)$ parameters are required by the string model to describe the hadronization.

In the cluster model, gluons are forced to split into quark–anti-quark pairs at longer distances (lower energy). All quark–anti-quark pairs are grouped into color singlet combinations with a distance scale that depends only on the evolution step, and not the hard process step of the Monte Carlo event generation. Hadrons are emitted from these universally pre-confined clusters via a series of two-body decays until only physical hadrons remain. The model has fewer parameters and naturally generates hadron transverse momenta. However, the decays of massive clusters lead to phenomenological problems such as predicting heavy baryon distributions which do not match data well.

Machine Learning (ML) techniques offer the possibility to build alternatives to the above two models of hadronization. Such ML models could be directly built from data and provide insights into the current phenomenological models. While ML techniques have recently entered into the development of event generators, through adaptive integration [15–20], parton showers [21–29], ML based fast detector or event simulations [28–55], and model parameter tuning [56, 57], the application of ML to the problem of hadronization as the final step in the Monte Carlo pipeline is entirely new, to the best of our knowledge. The present manuscript represents the first step toward building a full-fledged ML based hadronization framework.

In principle, Generative Adversarial Networks (GANs) [58], Variational Auto-Encoders (VAEs) [59], and Normalizing Flows (NF) [60] have demonstrated the ability of ML to generate convincing physical observables. In addition, conditional generative models give more flexibility and control of the output [61, 62]. Extending the ML techniques for hadronization faces three challenges: (i) producing sets of physical observables that vary in size (unlike a fixed number of pixels), ranging from $\mathcal{O}(1)$ to $\mathcal{O}(10^4)$; (ii) strictly conserving certain physical quantities, e.g., momentum and energy; and (iii) learning from limited training sets which only provide coarse-grain detail. In this paper, we present an architecture based on conditional sliced-Wasserstein autoencoders (cSWAE) [63, 64], that overcomes the above challenges. The resulting code, MLHAD, is publicly available, see Appendix A. We demonstrate the capabilities

of MLHAD by training it on specially prepared PYTHIA hadronization outputs with an explicit IR cut-off. To speed up the training we perform a transformation that captures the bulk of the energy dependence of the PYTHIA hadronization output. However, we also show that, if this transformation is not performed, the cSWAE can still reproduce the energy dependence and thus should be able to reproduce any additional energy dependence that may be present in the hadronization process realized in nature. We expect that the first version of the cSWAE architecture presented here can be upgraded to eventually be trained directly on data (details about further steps to achieve this can be found in section 4).

The paper is structured as follows. In Section 2 we introduce conditional sliced-Wasserstein autoencoders and describe how these can be used to reproduce the Lund string model of hadronization. In Section 3 we then compare the trained MLHAD models to the results of a simplified PYTHIA hadronization model. Section 4 contains our conclusions and a brief discussion of future directions. Appendix A contains details about the publicly accessible MLHAD code, while Appendix B gives further details on the sliced-Wasserstein distance.

## 2 Conditional SWAEs and hadronization

### 2.1 The simplified Lund string hadronization model

As the first step toward building a machine learning (ML) based simulator of hadronization, we create a ML architecture that is able to reproduce a somewhat simplified Lund string model for hadronization. Hadronization is the last step in the Monte Carlo simulation of the particle collision, and describes the creation of hadrons from quarks and gluons, a process that occurs at the nonperturbative scale of a few 100 MeV. The distributions of quarks and gluons at low scales is obtained using a parton shower simulation, which describes the emission of particles between the hard scale of the collisions, typically a few 100 GeV, down to low energies. In a Lund string model the quarks and gluons are thought of being connected by QCD color flux tubes, or strings, that carry significant amounts of energy, and shed it in the process of hadron creation. While there were already attempts to use ML to improve parton shower simulations [27,65–71], this manuscript represents the first attempt to use ML for hadronization. In both cases the physics is described by a Markov chain, however, for different reasons. The semi-classical evolution of a parton shower, where gluons and quarks are radiated in a Markov chain, can be justified in the small angle emission limit. The hadronization, on the other hand, can be represented as a Markov chain process because string fragmentations occur at causally disconnected points.

The physical process we want to describe is depicted in Fig. 1. It shows a $q_i\bar{q}_i$ fragmentation event in the center-of-mass frame in which the individual partons, each with flavor index $i$ and initial energy $E$, travel with equal and opposite momenta and are connected via a QCD string. String breaking produces a composite hadron $h \sim q_i\bar{q}_j$ and a new $q_j\bar{q}_i$-string system depicted in the lower part of Fig. 1.[1] The hadron $h$ is ejected with some energy and momentum $(E_h, \vec{p}_h)$, while the new string system has the energy and momentum $(2E - E_h, -\vec{p}_h)$, so that the total energy and momentum are conserved. The goal of our ML framework will be to properly describe the probabilities of emitting a hadron of given energy and momentum.

After boosting to the center-of-mass frame of the new string, one has essentially the same initial state, a quark–anti-quark pair going back to back connected by a string, but with reduced energy $E'$ and a different quark flavor composition. Such fragmentation events stack one after the other and form a fragmentation chain, one hadron emission at a time, until the

---

[1]The depiction in Fig. 1 is for a string breaking occurring on the quark side. The string breaking on the anti-quark side produces similarly a hadron with quark composition $h \sim q_j\bar{q}_i$, and the new $q_i\bar{q}_j$-string.

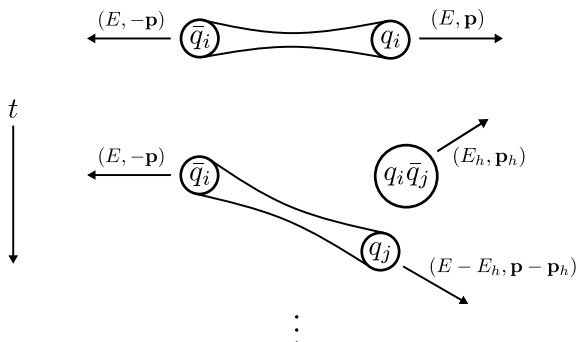

Figure 1: Schematic of a single fragmentation event, for an initial quark–anti-quark pair, $q_i\bar{q}_i$, into a hadron with quarks $q_i\bar{q}_j$ and new endpoints $\bar{q}_i q_j$.

entire energy of the initial two-parton system ($2E$) is converted into hadrons. The end of the string used for each splitting is chosen at random. Until relatively low string energies of a few GeV, the selection of flavor and the kinematics of the hadron emission are taken to be independent processes. In the final stages of hadronization, when the string energy is close to the nonperturbative scale, the two processes, on the other hand, become intertwined. To simplify the problem, we therefore terminate fragmentation events at a center-of-mass string energy $E_{\text{cut}} = 5$ GeV. We also consider a simplified string system which allows for $u$ and $d$ quarks as string ends, as well as their respective anti-quarks, and pions as final states.

Note that each step in the above hadronization chain is independent from the previous one. A successful hadronization simulator therefore takes as the input the string energy $E$ (i.e., the energy of one of the endpoint quarks in the center-of-mass frame) as well as its flavor composition, and gives the flavor and kinematics of the hadron after first emission, $(E_h, \vec{p}_h)$. Repeating the first emission generates the full hadronization chain. Since $E_h^2 = \vec{p}_h^2 + m_h^2$, where $m_h$ is the hadron mass, the kinematics of the emission are fully described by specifying $\vec{p}_h$ and flavor of the created hadron $h$. We orient the coordinate system such that the $z$ axis is along the direction of the initial string, while the $x$ and $y$ coordinates are perpendicular to it. The transverse components of the $\vec{p}_h$ vector are given by

$$p_x = p_T \cos\varphi, \qquad p_y = p_T \sin\varphi, \tag{1}$$

where $p_T \equiv \sqrt{p_x^2 + p_y^2}$ and $\varphi$ is the polar angle. The string breaking and hadron emission are assumed to be axially symmetric in PYTHIA, i.e., independent of $\varphi$, and thus the problem of simulating the hadronization event reduces to a two variable problem of generating the $p_z$ and $p_T$ distributions for the first emission.

A special feature of the hadronization event and the chosen kinematic variables is the ability to render the $p_z$ kinematic distributions independent of the initial parton energy, $E$, through a simple rescaling transformation

$$p_z' \equiv E_{\text{ref}} \frac{p}{E}, \tag{2}$$

where $E$ is the energy of the quark in the center of mass for the initial string, and $E_{\text{ref}}$ is a conveniently chosen reference energy that renders $p'$ dimensionful. In the rest of the paper we set $E_{\text{ref}} = 50$ GeV. The transformation of the $p_z$ distribution with respect to the initial parton energy $E$ can be seen in Fig. 2.

The fragmentation process implemented in PYTHIA is constructed in momentum space as an iterative walk through production vertices. To do so a stochastic variable termed the longitudinal momentum fraction $z$ is defined, describing the fraction of longitudinal momen-

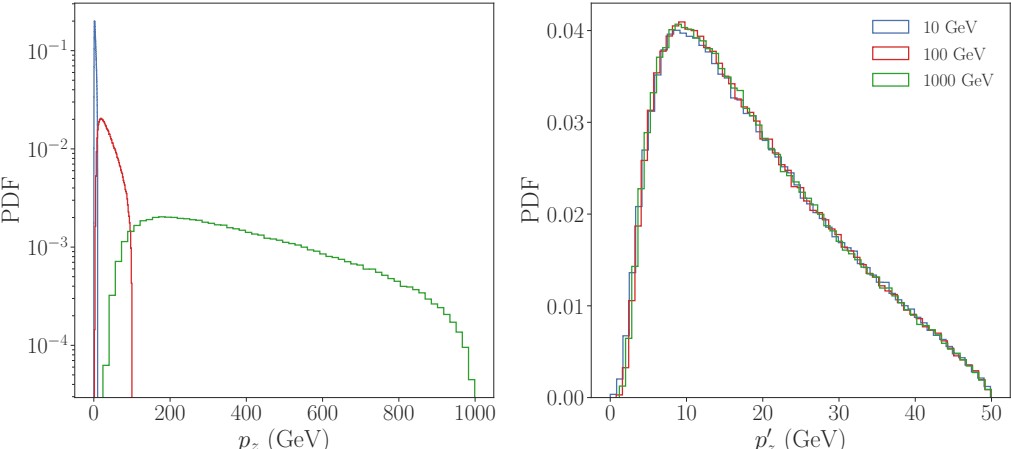

Figure 2: The $p_z$ distributions (left) and the rescaled $p'_z$, Eq. (2), distributions (right) from PYTHIA hadronization events for the first-hadron emission with initial parton energies $E = 10, 100, 1000$ GeV shown with blue, red, and green solid lines, respectively.

tum taken away by the emitted hadron.[2] Given the longitudinal momentum fraction, $p_z$ can straight-forwardly be obtained via the relation $z = (p_z + E_h)/2E$ where $2E$ is the total energy of the initial fragmenting system. The probability distribution $f(z)$ from which $z$ is sampled is called the *Lund left-right symmetric scaling function (also Lund sampling or fragmentation function)* and is given by

$$f(z) \propto \frac{(1-z)^a}{z} \exp\left(-b\frac{m_{h,T}^2}{z}\right),$$ (3)

where $m_{h,T}^2 \equiv m_h^2 + p_T^2$ is the transverse mass, and the normalization prefactor is omitted for clarity. The phenomenological parameters $a, b$ are chosen to match experimental data. The $p_T^2$ term in the transverse mass squared, $m_{h,T}^2$, captures the tunneling probability for a string breaking to occur away from the classical position of the string end, such that the additional energy required for the transverse momentum kick can be released from the string. It leads to a correlation between transverse and longitudinal distributions of hadron momenta (in the center-of-mass frame of the initial string), i.e., the average value of $z$ increases with increasing $p_T$. In the default implementation of the Lund model in PYTHIA, the hadron $p_T$ distribution is assumed to be Gaussian distributed, with average $\langle \vec{p}_T \rangle = 0$, and a width $\sigma_0 \sim \mathcal{O}(300\,\text{MeV})$, reflecting that its origin is an inherently quantum process occurring at the nonperturbative QCD scale.[3]

The above basic setup of the Lund model becomes more involved when full complexity of the experimental data needs to be explained. Most of the $\mathcal{O}(20)$ parameters that give more flexibility to the PYTHIA implementation of the Lund string model are related to the differences in hadronizations of the light quarks compared to the strange, $c$ and $b$ quarks. For instance, each quark flavor can in principle have a different $a$; in PYTHIA strange quarks are allowed to have different values of $a$ than for $u$ and $d$ quarks, while for heavier $c$ and $b$ quarks the Lund fragmentation is also allowed to be multiplied by an extra $z$-dependent factor with new flavor-dependent parameters. Similarly, the $p_T$ distributions can deviate from the Gaussian form. While this gives quite some flexibility to the hadronization model, it does have

---

[2]In Section 2.2, $z_i$ denote the latent-space variables. Despite similarity in notation there is no relation between the two variables.

[3]The configurable PYTHIA parameter name is `StringPT:sigma`.

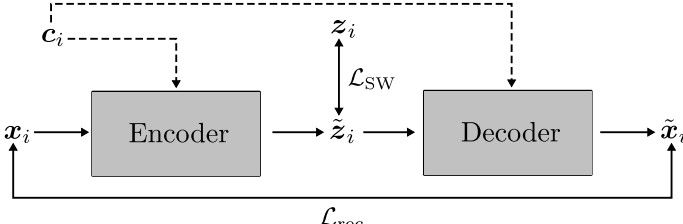

Figure 3: The cSWAE architecture for simulating hadronization. Inputs $x_i$ have condition $c_i$, which parametrizes the string energy. The decoder takes $\tilde{z}_i$ as inputs and generates the predicted hadron kinematics $\tilde{x}_i = \{\tilde{p}_{z,k}^{(i)}\}$. The sliced-Wasserstein-distance loss function, $\mathcal{L}_{SW}$, constrains the latent-space vectors $\tilde{z}_i$ to the target distribution $\tilde{z}_i \sim I(\tilde{z}_i, c_i)$. The reconstruction loss function, $\mathcal{L}_{rec}$, minimizes the difference between $x_i$ and $\tilde{x}_i$.

its own drawbacks. On one hand, the number of parameters to be tuned to data is already quite large. On the other hand, one may worry that the analytic form of the scaling function in Eq. (3), while well motivated, is not flexible enough, with higher order corrections in $z$ potentially becoming important, e.g., at low string energies. Generative ML models, such as the architecture that we introduce in the next section, can be used as effective tools to address both of these issues. For the purposes of this paper, we will not yet train our ML architecture on the physics data, but rather on the synthetic data generated by PYTHIA. However, we anticipate that the expressibility of the ML framework, which we demonstrate below, will allow for a better description of the physics data sensitive to hadronization than the Lund left-right symmetric scaling function in Eq. (3) does right now.

## 2.2 The cSWAE architecture

The ML model of hadronization used here is based on the conditional sliced-Wasserstein Autoencoder (cSWAE) [63, 64] (for an example of a use of SWAE architecture in particle physics simulations see [40]). The motivation for using cSWAE is two-fold, *i)* the flexibility of being able to use a wide variety of latent-space distributions and thus optimize the performance of the hadronization model, and *ii)* the ability to incorporate the energy dependence of hadronization through a two dimensional condition vector $c$. We expect the second feature to become most relevant once MLHAD is trained on experimental data, for which small breakings of the energy independence exhibited by the Monte Carlo generated $p_z'$ data, Fig. 2, may be anticipated. The main advantage of SWAEs over VAEs is the flexibility in the choice of the latent space distribution, which allows the user to choose any sampleable distribution as latent space distribution. This is achieved by introducing a sliced Wasserstein distance (i.e. an approximate of the real Wasserstein distance between the desired and the obtained latent space distributions) in the cost function, see Eq. (6) below. This is then added to the usual reconstruction loss estimate term in the cost function, see Eq. (5) below.

The schematic of the cSWAE architecture is given in Fig. 3. It has two parts, the encoder and the decoder:

The encoder $\phi$ takes as inputs the data vectors $x_i$ and labels $c_i$ and returns a latent-space vector $\tilde{z}_i = \phi(x_i, c_i)$. Depending on the value of $c_i$ the encoder will transform $x_i$ to different regions in the latent space, as shown in the graphical representation of Fig. 4. The dimension of the latent space, $d_z$, needed for the application to hadronization is anywhere from $d_z = 2$ to $d_z = 30$, see also Table 1. The latent-space vectors $\tilde{z}_i$ are trained to be distributed according to the target latent-space distribution, $\tilde{z}_i \sim I(\tilde{z}_i, c_i)$, which is ensured through the use of sliced-

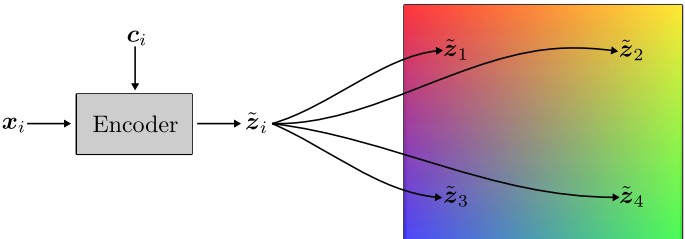

Figure 4: Illustration of the conditional vector $c_i = c(E_i)$ mapping the input data $x_i$ into different regions of the latent space, $\tilde{z}$.

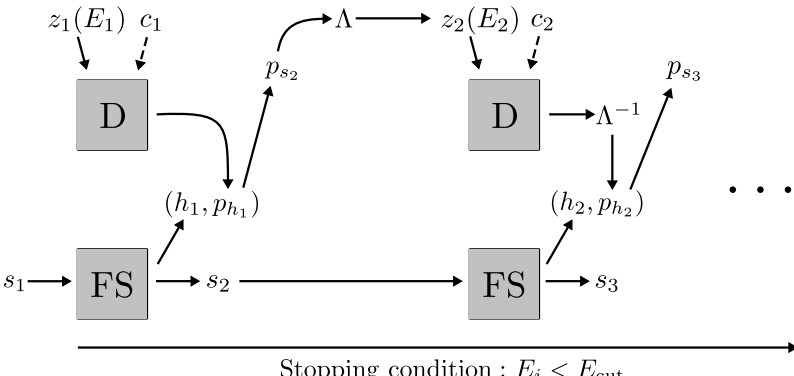

Figure 5: Illustration of MLHAD generating hadronization chains. Random variables $z_i$ are passed through the decoder D with condition vector $c_i$ to generate the hadron momentum, given the string energy $E_i$. A modified PYTHIA flavor selector FS, generates the new string flavor, $s_{i+1}$, and emitted hadron species, $h_i$. Before each emission, the string is boosted to its center-of-mass frame using a Lorentz transformation $\Lambda$.

Wasserstein distance, $SW_p$, in the loss function. In particular, the latent-space variable $\tilde{z}_i$ need not be normally distributed. We found that this feature translated to significant improvements in the performance of MLHAD. With cSWAE one can choose a custom probability distribution such that the encoding of the information about the first emission hadron kinematics leads to optimal results. This is the main practical difference between cSWAE and the conditional Variational Autoencoder (cVAE). The cVAE use KL-divergence in the loss function, which typically requires that the latent-space variables follow simple distributions, such as a normal distribution. The cSWAE uses instead the sliced-Wasserstein distance, $SW_p$, see Appendix B for more details. This gives the architecture significantly more flexibility, as one can choose the latent-space distributions to follow almost any distribution, as long as it is sampleable (in particular, the analytic form of $I(z, c_i)$ is not required to exist).

The decoder $\psi$ takes as inputs the condition vector $c_i$ and the latent-space vector $\tilde{z}_i$. It returns the reconstructed hadron kinematics $\tilde{x}_i = \psi(\phi(x_i, c_i))$, where $\tilde{x}_i$ is the $N_e$ dimensional vector consisting of sorted kinematic variables, either $p_{z,k}^{\prime(i)}$ or $p_{T,k}^{(i)}$. Through the minimization of the loss function [63]

$$\mathcal{L}(\psi, \phi) = \mathcal{L}_{\text{rec}} + \mathcal{L}_{\text{SW}}, \tag{4}$$

where

$$\mathcal{L}_{\text{rec}} = \frac{1}{N_{\text{tr}}} \sum_{i=1}^{N_{\text{tr}}} \left[ \frac{1}{Q} d_2^2(\boldsymbol{x}_i, \boldsymbol{\psi}(\boldsymbol{\phi}(\boldsymbol{x}_i, \boldsymbol{c}_i))) + d_1(\boldsymbol{x}_i, \boldsymbol{\psi}(\boldsymbol{\phi}(\boldsymbol{x}_i, \boldsymbol{c}_i))) \right], \tag{5}$$

$$\mathcal{L}_{\text{SW}} = \frac{\lambda}{L N_{\text{tr}}} \sum_{\ell=1}^{L} \sum_{i=1}^{N_{\text{tr}}} d_{\text{SW}}(\boldsymbol{\theta}_\ell \cdot \boldsymbol{z}_{[i]_\ell}, \boldsymbol{\theta}_\ell \cdot \boldsymbol{\phi}(\boldsymbol{x}_{[i]_\ell}, \boldsymbol{c}_i)), \tag{6}$$

with $\boldsymbol{z}_i \sim I(\boldsymbol{z}_i, \boldsymbol{c}_i)$, the training attempts to reproduce the training data distribution $\boldsymbol{x}_i$ with the generated data distribution $\tilde{\boldsymbol{x}}_i$, while the latent-space vectors $\tilde{\boldsymbol{z}}_i$ follow the desired target distribution $\tilde{\boldsymbol{z}}_i \sim I(\tilde{\boldsymbol{z}}_i, \boldsymbol{c}_i)$. The reconstruction loss $\mathcal{L}_{\text{rec}}$ is a measure of the differences between the input, $\boldsymbol{x}_i$, and generated kinematic vectors, $\tilde{\boldsymbol{x}}_i$. It is the sum of two terms for each of the 1D distributions that we consider,

$$d_2^2(\boldsymbol{x}_i, \boldsymbol{\psi}(\boldsymbol{\phi}(\boldsymbol{x}_i, \boldsymbol{c}_i))) = \begin{cases} \sum_k \left( p_{z,k}'^{(i)} - \tilde{p}_{z,k}'^{(i)} \right)^2, & \text{for } p_z' \text{ distributions}, \\ \sum_k \left( p_{T,k}^{(i)} - \tilde{p}_{T,k}^{(i)} \right)^2, & \text{for } p_T \text{ distributions}, \end{cases} \tag{7}$$

$$d_1(\boldsymbol{x}_i, \boldsymbol{\psi}(\boldsymbol{\phi}(\boldsymbol{x}_i, \boldsymbol{c}_i))) = \begin{cases} \sum_k \left| p_{z,k}'^{(i)} - \tilde{p}_{z,k}'^{(i)} \right|, & \text{for } p_z' \text{ distributions}, \\ \sum_k \left| p_{T,k}^{(i)} - \tilde{p}_{T,k}^{(i)} \right|, & \text{for } p_T \text{ distributions}, \end{cases} \tag{8}$$

where $p_{z,k}'^{(i)}$ and $p_{T,k}^{(i)}$ are the components of the training-dataset vectors $\boldsymbol{x}_i$, while $\tilde{p}_{z,k}'^{(i)}$ and $\tilde{p}_{T,k}^{(i)}$ are the components of the output vectors $\tilde{\boldsymbol{x}}_i$. For the relative weight between the two terms in $\mathcal{L}_{\text{rec}}$ we take $Q = 1\,\text{GeV}$. The two contributions of $\mathcal{L}_{\text{rec}}$ are sensitive to distinct scales allowing for fast convergence ($d_1$) and continual improvement ($d_2$) throughout training while also heavily penalizing outliers.

The second term in Eq. (4), $\mathcal{L}_{\text{SW}}$, is the implementation of the sliced-Wasserstein distance $SW_1$ between the distribution of latent-space vectors $\tilde{\boldsymbol{z}}_i$ created by the encoder, and the target latent-space distribution $I(\boldsymbol{z}_i, \boldsymbol{c}_i)$. The sliced-Wasserstein distance is the approximation of the true Wasserstein distance between the two distributions, and is smaller the closer the latent space distribution is to the desired one. The sliced-Wasserstein distance approximation becomes better and better the higher the number of 1D slices (or probes) of the distributions one uses. The advantage is that the computation of Wasserstein distances for 1D slices can be done very efficiently, leading to a significant speed up of the algorithm.

The computation of $SW_1$ is done as follows. The vectors $\boldsymbol{z}_i$ in Eq. (6) are randomly drawn from this target distribution, $\boldsymbol{z}_i \sim I(\boldsymbol{z}, \boldsymbol{c}_i)$. The scalar products with the unit vectors $\boldsymbol{\theta}_l$, defining the $L$ slices, give the one dimensional projections of the latent-space distributions, for which the Wasserstein distances, $W_1$, are straightforward to compute. They are given simply by the average sum of the distances between the sorted data points, see Appendix B for further details. Note that for one dimensional latent space $SW_1 = W_1$, and in the sum in Eq. (4) one can set $L = 1$.

## 2.3 Training

The input data to the encoder are $N_e$ PYTHIA generated first-hadron emissions for a fixed initial string energy $E_i = 50\,\text{GeV}$. In all of the numerical examples below we take $N_e = 100$, so that the input is an $N_e$ dimensional vector $\boldsymbol{x}_i$ of either $p_{z,k}'^{(i)}$ or $p_{T,k}^{(i)}$, $k = 1, \ldots, N_e$. That is, in this manuscript we apply cSWAE to the case where the $p_z'$ and $p_T$ distributions are uncorrelated and treat each of them separately. However, the architecture is flexible enough that correlated 2D or higher dimensional distributions could also be used as inputs.

The elements of the input vectors $\boldsymbol{x}_i$ are sorted, i.e., $p_{z,1}'^{(i)} \leq p_{z,2}'^{(i)} \leq \cdots \leq p_{z,N_e}'^{(i)}$ (and similarly for $p_{T,k}^{(i)}$).[4] The training dataset consists of $N_{\text{tr}}$ such $\boldsymbol{x}_i$ input vectors, $i = 1, \ldots, N_{\text{tr}}$, and $N_{\text{val}}$ $\boldsymbol{y}_j$ validation vectors, $j = 1, \ldots, N_{\text{val}}$, where typically $N_{\text{tr}}$ is taken to be $N_{\text{tr}} = \mathcal{O}(4000)$ and $N_{\text{val}}$ an order of magnitude smaller. To summarize, the training and validation datasets are created by generating $N \equiv N_e(N_{\text{tr}} + N_{\text{val}}) = 4 \times 10^5$ PYTHIA first hadron emission events. The emission data ($p_z$ or $p_T$) is then partitioned randomly into $N_{\text{tr}} + N_{\text{val}}$ vectors of length $N_e = 100$. Finally, the elements in each vector are sorted from least to greatest.

The string energy $E_i$, or equivalently mass in the center-of-mass frame, is converted to a unit condition vector $\boldsymbol{c}_i = (\bar{c}_i, 1 - \bar{c}_i)$ with $\bar{c}_i \in [0, 1]$ a floating point number such that

$$E_i = E_{\min}\bar{c}_i + E_{\max}(1 - \bar{c}_i), \qquad \text{and thus} \qquad \bar{c}_i = \frac{E_{\max} - E_i}{E_{\max} - E_{\min}}, \tag{9}$$

where $E_{\min}$ and $E_{\max}$ are the reference minimal and maximal energies. A good choice for $E_{\max}$ is the maximal partonic collision energy in the simulation, while $E_{\min}$ can be taken to be the IR cutoff $E_{\text{cut}}$.

In general, the cSWAE allows for the initial string energy $E_i$ of each $\boldsymbol{x}_i$ to be different (but the same for all the $N_e$ components of $\boldsymbol{x}_i$). For the PYTHIA generated events the kinematic variable $p_z$ can be made $E$ independent through the transformation in Eq. (2) and thus $E_i$ can be set to a constant value, $E_i = 50$ GeV. As a proof of principle we also show in Section 3.2 that cSWAE models can be trained on $E$-dependent $\boldsymbol{x}_i$.

The algorithm for training the cSWAE is as follows. Applying the encoder to the input data sample $\{\boldsymbol{x}_1, .., \boldsymbol{x}_{N_{\text{tr}}}\}$ gives the latent-space vectors $\{\tilde{\boldsymbol{z}}_1, .., \tilde{\boldsymbol{z}}_{N_{\text{tr}}}\}$. To compute the sliced-Wasserstein distance term, Eq. (6), the unit vectors $\{\boldsymbol{\theta}_1, .., \boldsymbol{\theta}_L\}$ are randomly sampled from the $(d_z - 1)$-dimensional unit sphere $\mathcal{S}^{d_z - 1}$, while the $N_{\text{tr}}$ latent-space vectors $\{\boldsymbol{z}_1, \ldots, \boldsymbol{z}_{N_{\text{tr}}}\}$ are sampled from the target distribution, $\boldsymbol{z}_i \sim I(\boldsymbol{z}_i, \boldsymbol{c}_i)$. For each $\boldsymbol{\theta}_\ell$, the scalar products $\boldsymbol{\theta}_\ell \cdot \tilde{\boldsymbol{z}}_i = \boldsymbol{\theta}_l \cdot \boldsymbol{\phi}(\boldsymbol{x}_i)$ and $\boldsymbol{\theta}_\ell \cdot \boldsymbol{z}_i$ are sorted in the following way. First the energy labels $c_i$ (and the corresponding $\tilde{\boldsymbol{z}}_i$, $\boldsymbol{z}_i$) are sorted into $N_c$ bins of increasing $c_i$ intervals with boundaries $\bar{c}_{[1]} < \bar{c}_{[2]} < \cdots < \bar{c}_{[N_c]}$. That is, the latent-space data are binned according to their energies, $E_i$, where the bins are chosen such that the distributions $I(\boldsymbol{z}_i, \boldsymbol{c}_i)$ do not have a large dependence on $c_i$ within the bin. The generated and target $I(\boldsymbol{z}_i, \boldsymbol{c}_i)$ distributions are then compared within each energy bin. This is achieved by first sorting the scalar products of $\tilde{\boldsymbol{z}}_i$ and $\boldsymbol{z}_i$ with $\boldsymbol{\theta}_\ell$ within each $c_i$ bin, and then combined into the lists $\{\boldsymbol{\theta}_\ell \cdot \tilde{\boldsymbol{z}}_{[1]_\ell}, \ldots, \boldsymbol{\theta}_\ell \cdot \tilde{\boldsymbol{z}}_{[N_{\text{tr}}]_\ell}\}$ and $\{\boldsymbol{\theta}_\ell \cdot \boldsymbol{z}_{[1]_\ell}, \ldots, \boldsymbol{\theta}_\ell \cdot \boldsymbol{z}_{[N_{\text{tr}}]_\ell}\}$, respectively. The SW loss function $\mathcal{L}_{\text{SW}}$ in Eq. (6) is then the average over the latent space distances between the two sorted lists,

$$d_{\text{SW}}(\boldsymbol{\theta}_\ell \cdot \boldsymbol{z}_{[i]_\ell}, \boldsymbol{\theta}_\ell \cdot \boldsymbol{\phi}(\boldsymbol{x}_{[i]_\ell})) = \left| \boldsymbol{\theta}_\ell \cdot \boldsymbol{z}_{[i]_\ell} - \boldsymbol{\theta}_\ell \cdot \boldsymbol{\phi}(\boldsymbol{x}_{[i]_\ell}) \right|, \tag{10}$$

averaged also over all the $L$ slices and multiplied by the relative weight prefactor $\lambda$. The final step in the algorithm is applying the decoder to $\tilde{\boldsymbol{z}}_i$, which gives $\{\tilde{\boldsymbol{x}}_1, \ldots, \tilde{\boldsymbol{x}}_{N_{\text{tr}}}\}$. The distances between the input dataset, $\{\boldsymbol{x}_1, .., \boldsymbol{x}_{N_{\text{tr}}}\}$, and the generated sets $\{\tilde{\boldsymbol{x}}_1, \ldots, \tilde{\boldsymbol{x}}_{N_{\text{tr}}}\}$ are then calculated using Eqs. (7) and (8), giving the reconstruction loss function $\mathcal{L}_{\text{rec}}$, Eq. (5). The decoder and encoder are updated in steps, trying to minimize the combined loss function, Eq. (4). Overfitting is avoided by monitoring the value of loss function when applied to the validation dataset, i.e., the loss function (4) with $\boldsymbol{x}_i \rightarrow \boldsymbol{y}_i$, $N_{\text{tr}} \rightarrow N_{\text{val}}$.

Fig. 5 illustrates how the trained MLHAD decoder is used, along with the PYTHIA flavor selector, to generate the hadronization chain. Note, the full PYTHIA flavor selector is not needed here, but is included to allow for subsequent development. The flavor selector takes as input the initial string flavor ID, $s_i$, and gives as the output the flavor ID of the emitted hadron, $h_i$,

---

[4]For 2D or higher dimensional problems the data would first be clustered in predefined 1D bins and then sorted within each bin.

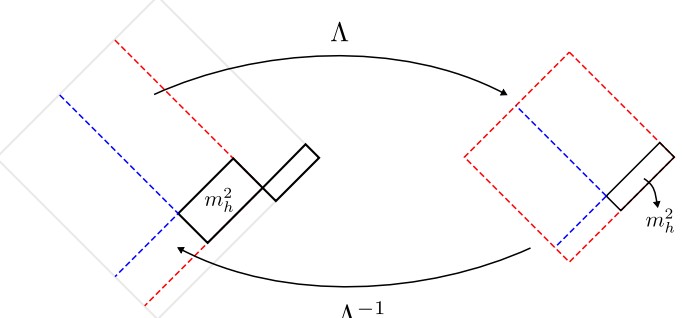

Figure 6: Illustration of Lorentz boosting ($\Lambda$) from the lab frame to the string center-of-mass frame. Red and blue lines are the string system's longitudinal momentum with the total area equal string system's longitudinal momentum $E + p_z$. Each box is a new string.

which also defines the flavor of the new string fragment, $s_{i+1}$. The MLHAD decoder takes as input the latent-space vector $z_i \sim I(z_i, c_i)$ sampled from the target distribution $I(z_i, c_i)$, where $c_i$ is the label encoding the center-of-mass energy of the string $s_i$, see Eq. (9). The MLHAD decoder returns the $N_e$-dimensional vector with a list of possible momenta for the emitted hadron, $\tilde{p}'^{(i)}_{z,k}$ (or $\tilde{p}^{(i)}_{T,k}$). We randomly choose one of these as the actual hadron kinematics, and modify accordingly the kinematics of the remaining string fragment, $s_{i+1}$, such that the energy and momentum are conserved. The emitted hadron is boosted to the lab frame, and added to the list of emitted hadrons, while the new string is boosted to its rest frame, see Fig. 6. Its center-of-mass energy defines the label $c_{i+1}$ used as the input in the decoder for the next hadron emission. These steps are repeated until the string energy in its rest frame reaches the IR cutoff energy $E_{\text{cut}}$.

We have implemented the cSWAE architecture described above using PYTORCH [72]. The code can be accessed via a public repository, see Appendix A for details.

## 3 Reproducing the simplified PYTHIA fragmentation model

To demonstrate the viability and capability of the cSWAE based machine learning algorithm implemented in MLHAD, we reproduce the PYTHIA hadronization outputs. We analyze a $q_i\bar{q}_i$ hadronization event in the center-of-mass frame in which the individual partons, each with flavor index $i$ and initial energy $E$, travel with equal and opposite momenta producing a string between them. After the string breaks this produces a new string and the first emission hadron, see Section 2.1 for more details.

While MLHAD treats all the hadron emissions on an equal footing, PYTHIA treats the first emission slightly differently; in the first emission $m_{T,h}$ in Eq. (3) is set to $m_h$ (i.e., $p_T = 0$), while for all subsequent emissions $p_x$ and $p_y$ are sampled from a normal distribution with a width $\sigma_0$ (we set this tunable PYTHIA parameter to $\sigma_0 = 0.335\,\text{GeV}$). Therefore, in training MLHAD we only aim to reproduce the PYTHIA output *on average*, which is in line with the physical limitations of the problem, since one cannot trace in nature each individual emission in the hadronization event.

Our model is trained on kinematic distributions for transformed variables, $p'_z$, $p_T$, Eq. (2), obtained from the PYTHIA first emission events. With a uniformly sampled polar angle $\varphi$ in the transverse plane, these kinematic variables then completely define the phase space of the system through Eqs. (1), (2). The MLHAD decoder is then used with a fixed shifted value transverse mass $m^2_{T,h} = m^2_h + \sigma^2$, with $\sigma = \sigma_0/\sqrt{2}$. This accounts for using only PYTHIA

Table 1: The cSWAE training configurations, where $\boldsymbol{x}$ is the input data, $\boldsymbol{z}$ the target latent-space distribution, $t$ the number of epochs, $d_z$ the dimension of the latent space, $\lambda$ the regularization parameter of the sliced-Wasserstein loss, and $L$ the number of latent space projections (slices).

| Variable $\boldsymbol{x}$ | Target $\boldsymbol{z}$ | $t$ (epochs) | $d_z$ | $\lambda$ | $L$ |
|---|---|---|---|---|---|
| | PYTHIA | 150 | 35 | 35 | 15 |
| $p_z'$ | Trapezoidal | 300 | 2 | 20 | 30 |
| | Triangular | 150 | 2 | 30 | 25 |
| | PYTHIA | 100 | 20 | 30 | 30 |
| $p_T$ | Skew-norm | 120 | 4 | 20 | 25 |
| | Triangular | 120 | 4 | 15 | 25 |

produced first emission data where $p_T = 0$ GeV. For flavor selection we rely on PYTHIA's probabilistic model, and limit ourselves to light quarks, $u$, $d$ and only pions as the final state hadrons.

The independence of the distributions from the initial parton energy, see Fig. 2, allows the cSWAE model to be trained on a dataset using an arbitrary initial parton energy, $E_{\text{ref}}$, while the outputs of cSWAE hadronization generator can be transformed accordingly to obtain the distributions for any desired initial energy, $E$, using Eq. 2. While in the PYTHIA output the complete energy dependence is already captured with the simple rescaling in Eq. (2) we do not expect this to be entirely true for actual physical hadronization events realized in nature, for which subleading deviations from the scaling law in Eq. (2) may be anticipated. In Section 3.2 we demonstrate that such corrections to the scaling law can be captured by the cSWAE architecture.

## 3.1 First emission trained models

The cSWAE trained models differ according to the target latent-space distribution, $I(\boldsymbol{z}, \boldsymbol{c})$, the dimension of the latent space $d_z$, training time $t$ (epochs), the value of the sliced-Wasserstein regularization parameter $\lambda$, and the number of slices $L$, as shown in Table 1. In all the cases we fix the string energy to be $E = 50$ GeV. The first emissions for other string energies can be obtained by inverting the rescaling of the $p_z'$ distributions in Eq. (2), while $p_T$ distributions do not scale with $E$, although this is an assumption of the PYTHIA model. For PYTHIA generated $p_z'$ data we use the transverse pion mass $m_{T,\pi}^2 = m_\pi^2 + \sigma^2$, instead of the actual pion mass. Because of the different treatment of first and subsequent hadron emissions in PYTHIA, this choice for a pion mass will then reproduce the average PYTHIA hadronization results for full hadronization chains, as discussed at the beginning of Section 3 and shown explicitly in Section 3.3 below.

A key feature of the SWAE algorithm and the sliced-Wasserstein loss is the ability to 'push' the encoded latent space towards a target latent-space distribution. The choice of target distribution affects the total training time and the speed of kinematic data generation. Choosing a target latent-space distribution which is similar to the training data set distribution generally requires a fewer number of epochs to train the model to a specified accuracy compared to a target latent space which is dissimilar. This may come at a cost during the generation of kinematic data for hadronization events due to the generation of a large number of random variables obeying potentially complex probability distributions.

We demonstrate this flexibility by training with multiple target latent-space distributions, see Fig. 7. A total of six models are trained, three for each kinematic variable $p_z'$ and $p_T$, with the results shown in Figs. 8 and 9. Of the three models in each kinematic variable, one model

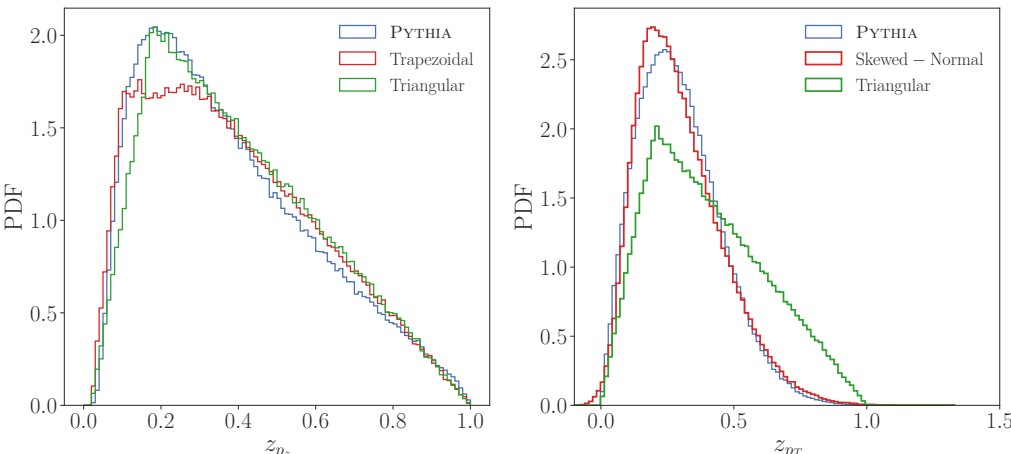

Figure 7: Three choices for latent-space target distributions $I(\boldsymbol{z}, \boldsymbol{c})$ for $p'_z$ inputs (left) and for $p_T$ inputs (right). See Appendix C for more details.

is trained using a target latent-space distribution equivalent to the training set distribution, i.e., the PYTHIA generated distribution of $p'_z$ or $p_T$. The other two trained models have target latent-space distributions that are distinctly different from the training set distributions. For $p'_z$ we choose trapezoidal and triangular target latent distributions and for $p_T$ we choose a skewed normal and triangular target latent-space distributions. The latent-space distributions are shown in Fig. 7, while their analytic forms can be found in Appendix C. Regardless of the choice of the latent-space distribution, the trained and the target (prior) data distributions are in good agreement.

The dimension of the latent space is a tunable discrete hyperparameter, taking values $d_z \in [2, 35]$, see the fourth column in Table 1. The regularization parameter $\lambda$ controls the magnitude of the sliced-Wasserstein loss and determines its relative weight in the total loss function, see Eq. (4). In practice, the regularization parameter determines how closely the encoded latent-space distribution will agree with the chosen target latent-space distribution, $I(\boldsymbol{z}, \boldsymbol{c})$. In our trained models the regularization parameter in the loss function Eq. (4) takes values $\lambda \in [15, 35]$, as listed in the fifth column in Table 1. Larger values are chosen in models where the target latent-space distribution is similar to the training distribution. Large values of $\lambda$ effectively reduce the size of the explored manifold which maps decoder weight-configurations to values of the loss function (if we think of the decoder as a partition function and the loss function as a functional, large values of $\lambda$ place the decoder near a saddle-point configuration). This improves the convergence to the minimum of $\mathcal{L}_{\text{rec}}$, resulting in shorter training times. This can also be explained by describing the correlation between the minimization of $\mathcal{L}_{\text{SW}}$ and $\mathcal{L}_{\text{rec}}$.

The number of slices or projections used in the sliced-Wasserstein loss is also a tunable hyperparameter taking values $L \in [15, 30]$, as listed in the last column in Table 1. Each model uses the kinematic data generated from $N = 4 \times 10^5$ first emission events partitioned into $N/N_e = 4000$ $N_e$-dimensional vectors, where 80% of the data is used as the training and 20% as the validation set. We use an initial learning rate value of $10^{-3}$ and utilize PYTORCH's dynamic learning-rate scheduler to reduce the learning rate according to the plateaus of the loss function during training.

## 3.2 Labels and $E$ dependent distributions

The trained models for the first-hadron emission presented in the previous section were all obtained for a fixed initial string energy, $E$. To reproduce the PYTHIA model for the first-

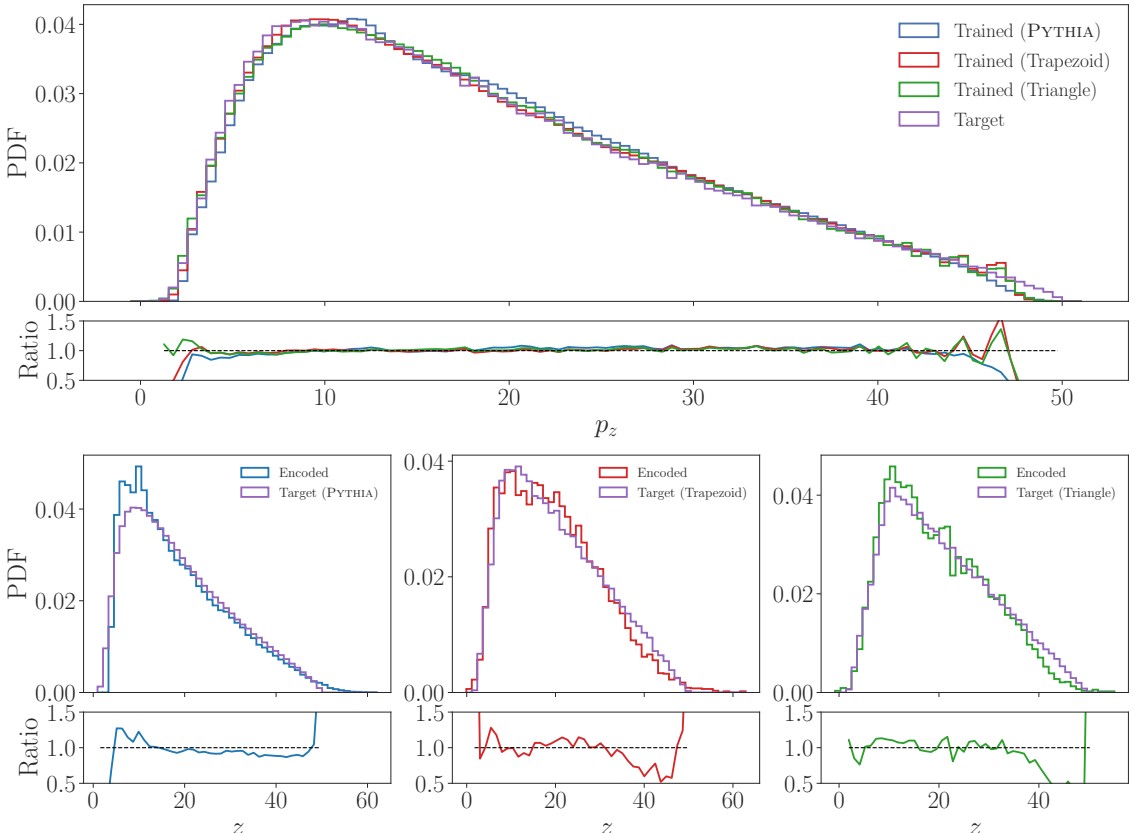

Figure 8: Top: MLHAD generated $p_z$ distributions for first-hadron emission from a string with an energy $E = 50$ GeV, using three different latent-space distributions, PYTHIA (blue), trapezoidal (red), and triangular (green), compared to the PYTHIA generated target distribution (purple), as well as the ratios of MLHAD generated to PYTHIA generated distributions. Bottom: comparison of the trained and target latent-space distributions for the three cases.

hadron emissions (for string fragments with energies above $E_{\text{cut}}$) this is all that is required. The $p_z'$ distributions for any string energy can be obtained from the reference value of $E = 50$ GeV that we used in the training by performing the rescaling, cf. Eq. (2) and Fig. 2. The $p_T$ distributions for first emissions, on the other hand, are independent of the initial string energy.

However, the above scaling behaviors are not expected to be exact in nature. For one, at lower string energies the approximations in deriving the string Lund model are likely to fail - the quarks are not massless, and there may be couplings between $p_T$ and $m_h$ that are not captured by the simple transverse mass tunneling ansatz, Eq. (3). Furthermore, the origin of $p_T$ distributions for first emissions is purely non-perturbative in nature, and thus the $E$ independence of $p_T$ distribution assumed in PYTHIA is not rooted in first principles.

The MLHAD architecture is flexible enough to allow for the dependence of first emissions on the string energy, $E$. This is achieved by training the conditional SWAE on label-dependent datasets, which we demonstrate next. The training proceeds in a similar way as in the previous section, but now on a dataset comprising of first-hadron emissions for four distinct string energies, $E = \{5, 30, 700, 1000\}$ GeV.[5] Each $x_i$ input vector is therefore accompanied by one of the four discrete values for the two-dimensional vectors $\boldsymbol{c}_i = (1 - c_i, c_i)$ encoding the string

---

[5]One could also have used emission data for continuous values of $E$, but binned finely enough in string energy values. We choose discrete string energies to demonstrate clearly that the cSWAE decoder can interpolate between the input labels.

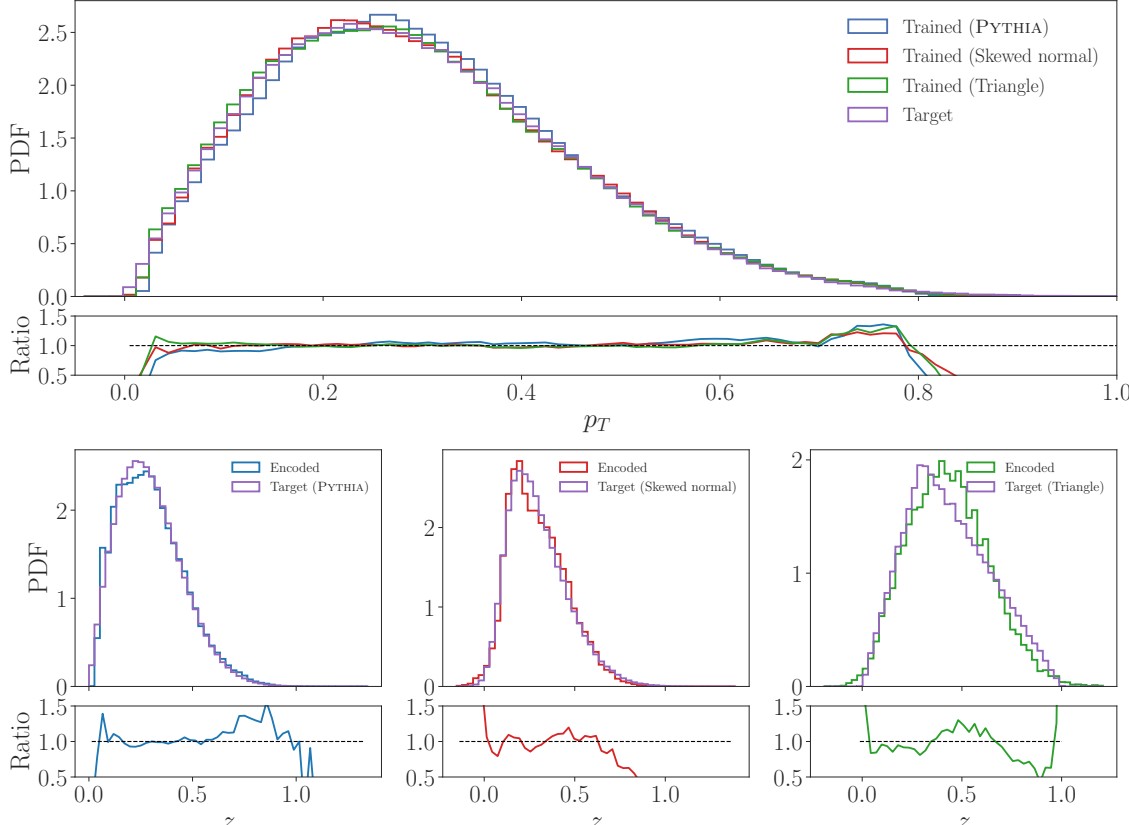

Figure 9: Top: MLHAD generated $p_T$ distributions for first-hadron emission using three different latent-space distributions, PYTHIA (blue), skewed-normal (red), and triangular (green), compared to the PYTHIA generated target distribution (purple), as well as the ratios of MLHAD generated to PYTHIA generated distributions. Bottom: comparison of the trained and target latent-space distributions for the three cases.

energy through the label $c_i$ as defined in Eq. (9), taking $E_{min} = 5$ GeV and $E_{max} = 1000$ GeV.

The decoder in the trained cSWAE was then used to generate the first-hadron emissions at a different set of string energies, $E = \{100, 200, 300, 400, 500\}$ GeV. Importantly, because the conditional vector is not discrete but rather depends on a continuous parameter defined between the minimum and maximum energies $(E_{min}, E_{max})$ the trained decoder is able to interpolate between labels (ones which the decoder has not trained on explicitly, see Fig. 4) and rescale the kinematic distributions accordingly. This considerably increases the flexibility of generating training datasets as the user is able to choose the number of interpolation points which the model can use as anchors in generating data with a unique energy label. The comparison of MLHAD and PYTHIA generated $p_z$ distributions for the first-hadron emissions is shown in Fig. 10, demonstrating that MLHAD reproduces faithfully the PYTHIA results.

## 3.3 Hadronization chain

As shown in the previous subsections the cSWAE trained models in MLHAD are able to accurately reproduce PYTHIA's first emission kinematics for a hadronized $q\bar{q}$ system in the center-of-mass frame of the string. In this section we show how well the MLHAD decoder reproduces the full PYTHIA hadronization event. The implementation can be summarized as follows: from the initial string system, one string end is chosen randomly, while PYTHIA flavor selector is used to determine the flavor ID of the emitted hadron. Given the energy of the initial string end

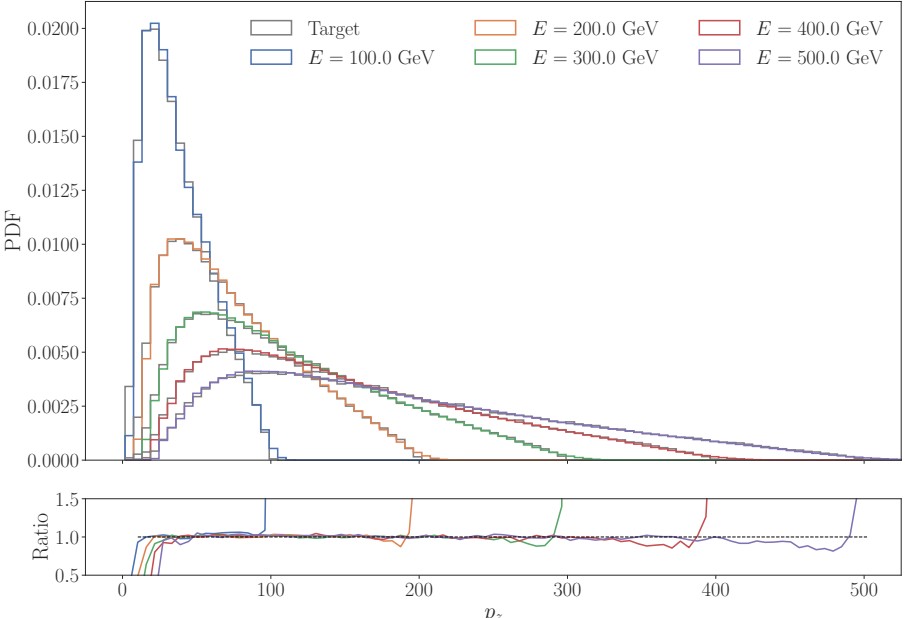

Figure 10: MLHAD generated $p_z$ distributions using the cSWAE model trained on data with string energies different from training and compared with PYTHIA (black).

in the center-of-mass frame, $p_z'$ and $p_T$ are sampled using the corresponding cSWAE models. The $p_z'$ and $p_T$ of the emitted hadron are transformed to $p_x, p_y, p_z$ variables using Eqs. (1) and (2), and boosted to the lab frame. The string fragment is boosted to its center-of-mass frame, see Fig. 6, after which one repeats the hadron emission process until the string energy in the center of mass of the remaining string fragment falls below the IR cutoff, $E_{\text{cut}}$. The implemented fragmentation chain architecture is illustrated in Fig. 5.

Fig. 11 shows a comparison between the hadronization chain multiplicities obtained by PYTHIA (blue) and by the MLHAD model trained on first emission data (red). In both cases, starting from the initial string energy of $E = 50$ GeV, on average 9.1 hadron emissions occur before the string fragment energy drops below the cutoff energy, $E_{\text{cut}} = 5$ GeV. The MLHAD decoder also reproduces well the distribution of hadronization chain multiplicities. Only a few hadronization events result in just a few hadrons, a bulk of hadronization events contain between 7 to 13 hadrons, and both hadronization chain generators feature a tail of rather long hadronization chains. The differences between the PYTHIA and MLHAD hadron multiplicity distributions are in most cases at the level of $5 - 10\%$, where the largest deviations occur for hadronization events with just a few hadron emissions. This is expected, given that PYTHIA and MLHAD models of hadronization differ in the treatment of the very first emission, see the discussion at the beginning of Section 3.

In Fig. 12 we also show the comparison of the average multiplicity of the hadronization chain as a function of the initial parton energy, obtained either with PYTHIA (blue solid line) or with MLHAD (red). We see that MLHAD is able to reproduce the PYTHIA fragmentation chain length averages, and in particular also give the expected $\log E$ dependence of the average number of produced hadrons. For each energy the multiplicity distributions also match well, which we checked explicitly, while in the figure we only show the result for MLHAD to guide the eye (red density plot). The density plot scan was performed by randomly choosing an initial parton energy $E$ between 20 GeV-1000 GeV and binning each fragmentation chain length with a parton energy resolution of 22 GeV and chain length resolution of 1.7 hadrons for a total of $2 \times 10^4$ fragmentation events. The minimal initial string energy was chosen to be 20 GeV such that it is still well above the imposed hadron emission cut $E_{\text{cut}} = 5$ GeV.

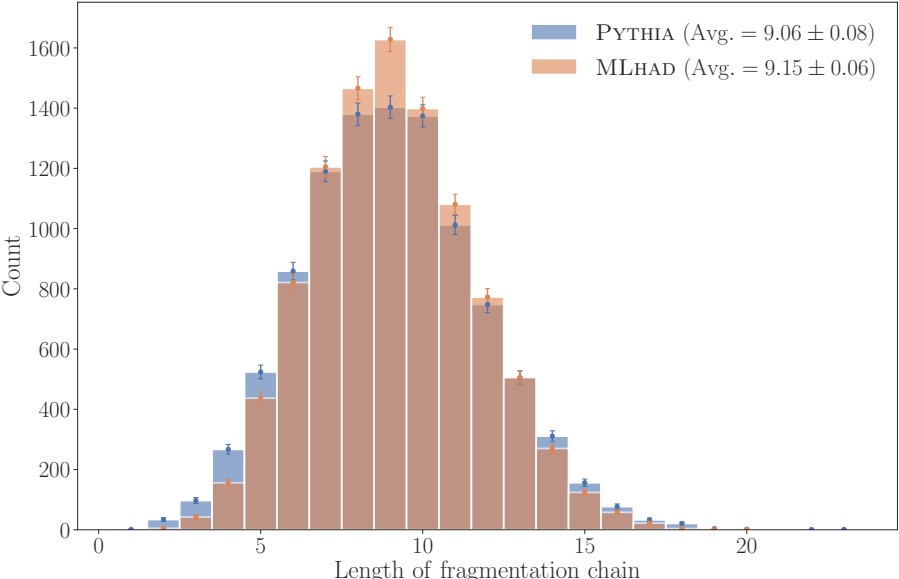

Figure 11: Comparison of the number of hadrons produced in the fragmentation chain of a single string for a sample of $10^4$ strings, compared between PYTHIA (blue) and MLHAD (red) generated hadronization events.

## 4 Conclusion and Outlook

The cSWAE architecture that was developed in this work appears to be well suited for modeling the nonperturbative process of hadronization – the creation of hadrons from the energy stored in the string connecting a $q\bar{q}$ pair. We have demonstrated this by training the MLHAD hadronization models to a simplified version of PYTHIA hadronization, limited to only light quark flavor endings of the string, and allowing only for pions to be the final-state hadrons. Furthermore, we utilized the scaling properties of the PYTHIA hadronization model that simplified the cSWAE training, requiring training at just a single string energy. Even so, the results shown in Figs. 8, 9 and 11 are very encouraging. The PYTHIA first-hadron emission distributions at a fixed string energy, Fig. 8, 9, are faithfully reproduced by the MLHAD decoder, as are the hadron multiplicities for full hadronization chains, Fig. 11.

The cSWAE architecture also has enough built in flexibility that it should be possible to extend the MLHAD model to handle all possible string flavors and kinematics. We have already shown that the inclusion of a label allows for an interpolation of the hadronization models to different string energies, see Fig. 10. This should then also allow to extend the MLHAD models below the string energy cut of 5 GeV that we imposed in this preliminary exploration. Similarly, the conditional label could be used for MLHAD to handle the generation of hadron flavors, including possible kinematic dependencies. The MLHAD architecture should also allow us to model any correlations between $p_z$ and $p_T$ distributions of the emitted hadrons, if these are present in data, even though currently we used the absence of such correlations in PYTHIA generated data to simplify the training of MLHAD models. Another important feature that we anticipate to be particularly important once MLHAD is trained directly on experimental data, is the flexibility in the choice of the latent-space distributions, making it easier to adapt to any possible features not captured by the rather constrained form of the Lund fragmentation function underlying the hadronization implementation in PYTHIA. Finally, some of the planned extensions of the MLHAD hadronization framework may require more thought, most notably how to best model the hadronization of baryons and include gluons.

While in this paper the training of MLHAD was performed on the first hadron emissions

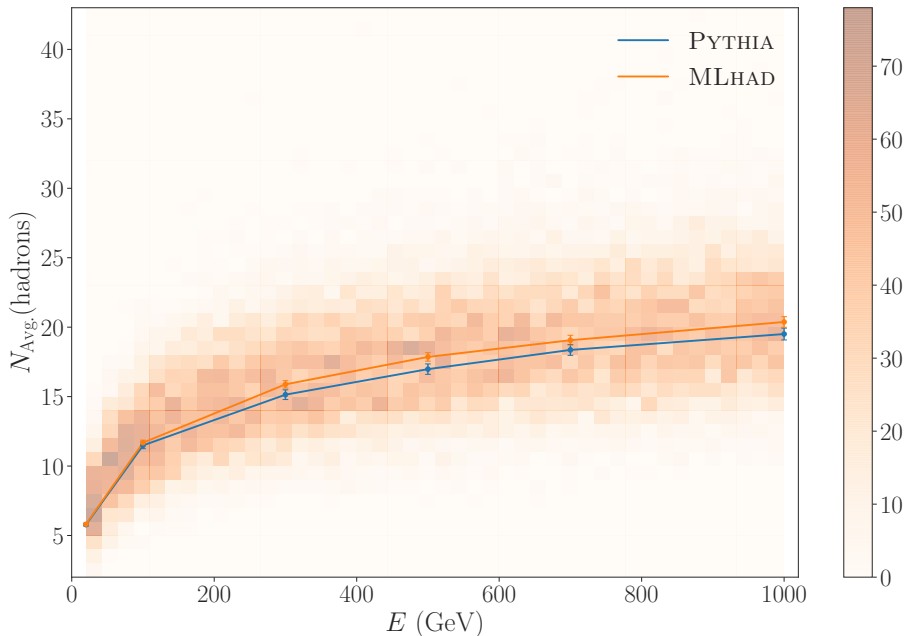

Figure 12: Comparison of the average number of hadrons produced in the fragmentation chain of a single string as a function of the initial parton energy $E$ ($E_{string} = 2E$), produced using PYTHIA (blue) and MLHAD (red). The density plot shows the multiplicity distributions obtained with MLHAD for $2 \times 10^4$ fragmentation chains.

in the PYTHIA output, such training will not be possible when using real experimental data, since such information is physically not possible to extract directly from data. Instead, the training will need to be performed on the physically accessible observables constructed from particle flows measured either in $e^+e^-$ or $pp$ collisions with two, three or more jets in the final state. We anticipate that this is where the machine learning approach to hadronization will prove most useful — capturing the many observables in principle available in the data, such as hadron multiplicities, angular separations and momentum distributions for various hadrons (see [73–78] for a selection of potentially useful observables). While many of these observables are not currently available in the literature, open-data efforts by a number of collaborations have or will make access possible. This data-collection is tedious when performed through human intervention and is a problem that calls for a machine learning based optimization. We believe that the presented MLHAD cSWAE architecture is well suited to achieve this next step, and we are in the process of building a pipeline to perform training of MLHAD on actual data. In addition different generative models like Normalizing Flows will be explored, which provide a tractable probability distribution function.

# Acknowledgments

We thank Jared Evans for collaboration in the initial stages of this work, and Stephen Mrenna, Manuel Szewc, and Mike Williams for useful comments on the manuscript.

**Funding information.** AY, JZ, and TM acknowledge support in part by the DOE grant de-sc0011784 and NSF OAC-2103889. PI is supported in part by NSF OAC-2103889.

# A  Public code `MLhad_v0.1`

The public code may be accessed through https://gitlab.com/uchep/mlhad. The public directory includes example files allowing the user to train and implement cSWAE models in full fragmentation chains. The programs are written in PYTHON and extensively use the PYTHIA, PYTORCH and SCIKIT-LEARN libraries. Installation instructions can be found on the respective installation pages for each library.

The provided programs can be split into two categories: training cSWAE models and generating hadronization events. The latter relies on the former. However, we have also provided pre-trained models such that the user can generate hadronization events without explicitly training a model.

Training a unique model configuration can be done by modifying the files `pT_SWAE.py`, `pz_SWAE.py`, or `pz_cSWAE.py`. The SWAE programs contain examples of label-independent training, while the cSWAE program provides an example of label-dependent training. The model hyperparameters and target latent distribution described in Section 2 have been set to default values to provide a reasonable starting configuration but may be modified. Label independent kinematic training datasets for $p_z$ and $p_T$ have been provided as well as a label-dependent $p_z$ dataset.

Full hadronization events use the trained model decoder to generate hadronic kinematics. An example of generating this kinematic data from SWAE trained model decoders can be found in `model_pxpypz.py`. The setup of our modified fragmentation chain which utilizes these kinematics can be seen in `frag_chain.py`.

# B  Sliced Wasserstein distance

In this appendix we give a short overview of the Wasserstein distance and the sliced-Wasserstein distance.

**The Wasserstein distance.**  The Earth mover's distance or the Wasserstein distance gives a measure of how different two distributions are, given a metric space $\Omega$ and a space of Borel probability measures $\mathcal{P}(\Omega)$ on $\Omega$. The $p$-Wasserstein distance $W_p(\mu, \nu)$ between any two probability measures $\mu \in \mathcal{P}(X)$ and $\nu \in \mathcal{P}(Y)$ is [79]

$$W_p(\mu, \nu) := \left( \inf_{\pi \in \Pi(\mu, \nu)} \int_X c(x, y) d\pi(x, y) \right)^{\frac{1}{p}}, \tag{11}$$

where $c(x, y)$ is the cost function, $\Pi(\mu, \nu)$ is the set of all transportation plans, with $\pi \in \Pi(\mu, \nu)$, while $p \in [1, \infty)$. The distance $W_1$ is also commonly called the Kantorovich-Rubinstein distance.

If $\mu$ and $\nu$ are one-dimensional measures, the Wasserstein distance has a closed-form expression

$$W_p(\mu, \nu) = \left( \int_0^1 |F_\mu^{-1}(z) - F_\nu^{-1}(z)|^p dz \right)^{1/p}, \tag{12}$$

where $F_{\mu(\nu)}(x) = \int_{-\infty}^x I_{\mu(\nu)}(\tau) d\tau$ are the cumulative distribution functions, with $I_\mu$ and $I_\nu$ the probability density functions for the measures $\mu$ and $\nu$, respectively. The $W_p(\mu, \nu)$ for the one dimensional case can therefore be calculated by simply sorting the samples from the two distributions and calculating the average cost.

**Radon transform and the sliced-Wasserstein distance.** An approximate value for the Wasserstein distance $W_p$ between two higher dimensional distributions on $X = \mathcal{R}^d$ can be obtained efficiently from a set of projections to one-dimensional distributions, since for each of these one can use the closed form of Eq. (12). The projection from the higher dimensional distribution to the one-dimensional representation is done by the Radon transform.

The $d$-dimensional Radon transform $R$ maps a function $I \in L^1(\mathcal{R}^d)$ to [80]

$$RI(t, \theta) = \int_{\mathcal{R}^d} |I(x)| \delta(t - \langle x, \theta \rangle) dx, \tag{13}$$

with $(t, \theta) \in \mathcal{R} \times \mathcal{S}^{d-1}$, where $\mathcal{S}^{d-1}$ is the unit sphere in $\mathcal{R}^d$, $\delta(\cdot)$ is the delta function and $\langle , \rangle$ is the Euclidean scalar product. For a fixed direction $\theta$ the Radon transform $RI_\mu(\cdot, \theta)$ therefore gives a one dimensional marginal distribution of $I_\mu$ that is obtained by integrating $I_\mu$ over the hyperplane orthogonal to $\theta$.

The sliced-Wasserstein distance $SW_p(I_\mu, I_\nu)$ between $I_\mu$ and $I_\nu$ is defined as

$$SW_p(I_\mu, I_\nu) = \left( \int_{\mathcal{S}^{d-1}} W_p(RI_\mu(\cdot, \theta), RI_\nu(\cdot, \theta)d\theta \right)^{\frac{1}{p}}. \tag{14}$$

The Wasserstein distance between each of the one dimensional projections (slicings) $RI_\mu(\cdot, \theta)$ and $RI_\nu(\cdot, \theta)$ is obtained straightforwardly using the closed form result of Eq. (12). The integral over the unit sphere vectors $\theta$ probes all the possible slicings. Furthermore, $SW_p(I_\mu, I_\nu)$ approximates $W_p(I_\mu, I_\nu)$ "well enough" [81].

The integration in Eq. (14) over the unit sphere in $\mathcal{R}^d$ can be estimated using a Monte Carlo integration that draws samples $\{\theta_l\}$ from the uniform distribution on $\mathcal{S}^{d-1}$, which substitutes a finite sample average for the integral [82],

$$SW_p(I_\mu, I_\nu) \approx \left( \frac{1}{L} \sum_{l=1}^{L} W_p(RI_\mu(\cdot, \theta_l), RI_\nu(\cdot, \theta_l)) \right)^{\frac{1}{p}}, \tag{15}$$

where $L$ is the number of projections (slicings). With this result, the sliced-Wasserstein distance is obtained by solving a finite number of one-dimensional optimal transport problems, each of which has a closed-form solution. Furthermore, the sliced-Wasserstein distance approximates well the Wasserstein distance and thus can be used as a useful discriminator for the similarity of distributions. More details can be found in [82] and [63].

## C  Latent distributions

The analytic forms of the latent target distributions used in the training of cSWAE in Section 3.1 are

$$I_{\text{tri.}}(z; a, b, c) = \begin{cases} \dfrac{2(z-a)}{(b-a)(c-a)}, & a \le z \le c, \\ \dfrac{2(b-z)}{(b-a)(b-c)}, & c < z \le b, \end{cases} \tag{16}$$

for the triangular distribution, and

$$I_{\text{trap.}}(z; a, b, c, d) = \begin{cases} \dfrac{2}{d+c-a-b} \dfrac{z-a}{b-a}, & a \le z < b, \\ \dfrac{2}{d+c-a-b}, & b \le z < c, \\ \dfrac{2}{d+c-a-b} \dfrac{d-z}{d-c}, & c \le z \le d, \end{cases} \tag{17}$$

Table 2: The $p_z'$ and $p_T$ latent-space distribution parameters.

| Variable $x$ | Target $z$ | $a$ | $b$ | $c$ | $d$ |
|---|---|---|---|---|---|
| $p_z'$ | Trapezoidal | $0.04E$ | $0.16E$ | $0.24E$ | $E$ |
| | Triangular | $0.04E$ | $0.2E$ | $E$ | – |
| $p_T$ | Triangular | $0.0$ | $0.3$ | $1.0$ | – |

for the trapezoidal distribution. For a given initial parton energy $E$ the choices of parameters $a, b, c, d$ can be seen in Table 2. The target latent-space distributions are then given by

$$I_{\text{tri.}}(\boldsymbol{z}, \boldsymbol{c}) = \prod_{k=1}^{N_e} I_{\text{tri.}}(z_k; a, b, c), \qquad I_{\text{trap.}}(\boldsymbol{z}, \boldsymbol{c}) = \prod_{k=1}^{N_e} I_{\text{trap}}(z_k; a, b, c, d), \tag{18}$$

that is we take the same values of $a, b, c, d$ parameters for all $d_z$ latent dimensions.

The normal and skewed-normal distributions are given by

$$I_{\text{Gauss}}(z; \mu, \sigma) = \frac{1}{\sigma\sqrt{2\pi}} \exp\left(-\frac{(z-\mu)^2}{2\sigma^2}\right), \tag{19}$$

$$I_{\text{Skew-Gauss}}(z; \mu, \sigma, \alpha) = 2 I_{\text{Gauss}}(z; \mu, \sigma) \Phi\left(\frac{\alpha(z-\mu)}{\sigma}\right), \tag{20}$$

respectively, where

$$\Phi(x) = \frac{1}{\sqrt{2\pi}} \int_{-\infty}^{x} e^{-t^2/2} dt. \tag{21}$$

The $\mu$, $\sigma$, and $\alpha$ are the fit parameters corresponding to the mean, standard deviation, and skewness of the distribution, respectively. As in Eq. (18) the $d_z$ dimensional latent-space distributions are products of one dimensional ones with the same $\mu, \sigma, \alpha$ parameters. For $p_T$ we have $\mu = 0.099$, $\sigma = 0.257$, and $\alpha = 4.259$.

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
