# Peer review of "Modeling Hadronization using Machine Learning"

_SciPost Physics, doi:SciPost Phys. 14, 027 (2023)_

## Round 1 · Referee Report · Tilman Plehn (Referee 1) · 2022-5-27

Strengths

Someone has to tackle hadronization with ML-methods atsome point, so thank you for this nice study!

Weaknesses

Very nice idea and paper, but the authors do need to work on the presentation and on putting their work into context in a few different ways.

Report

Thank you for the nice work and the nice paper. I have a few comments on the content and the readability, but on the whole the paper should definitely be published.

- in the introduction, the authors are missing the most modern generative networks based on normalizing flows. I also do not understand the point (iii) about limiting training data and coarse-grain detail being a case for a VAE;
- also, please introduce the concept of conditional generative networks in the introduction, there are examples for conditional GANs and conditional INNs from my group, and I am sure there are more;
- I appreciate the nice physics discussion in Sec.2.1, any chance you could already introduce the ML-task there? Is is just to learn Eq.(3), in essence? This might be a good point to go through ML-approaches to parton showers and compare the ways one can train a network on a simple Markov problem;
- Looking at Eq.(3), as a non-expert I do not really see where this formula lacks expressivity and needs to be modelled by a NN, could you please explain this a little?
- Section 2.2 is really hard to understand. The problem might be that it mixes an introduction of the cSWAE with the physics problem and the training data. Please separate these aspects, introduce the WAE, the conditional aspects, the physics problem, and the detailed implementation and training dataset. It probably means just reshuffling paragraphs and sentences a little, but it would make things much clearer;
- in Eq.(6), please explain the mix of L1 and L2 norms, this looks interesting for a broader audience;
- for now, I am not sure I buy the argument with the custom distributions in latent space. The kinematical patterns I see should be easily mapped to a Gaussian. Why is my intuition wrong there? And what does this mean quantitatively? Related to this question, what am I learning from Figs.8 and 9? I am sorry, but I am missing the point there;
- in Tab.1, please explain what d_z, lambda, L mean or where to find the definitions;
- Figs.11 and 12 are very convincing as a bottom line, but this is also not yet an ML-hadronization model. What are the next steps after this proof of concept? Again, as a non-expert I am confused by the focus on the first emission. What happens if I train a network on all the second emissions? Please provide some more context for a broader audience;
- the references are a little weird the way they are picked. Some mix of generative networks and networks used in generators. From my group, I would also have expected the original GAN paper rather than [22]. And maybe something on conditional networks. Maybe some detector simulations, like Caloflow, and definitely the generative shower models? For an overview, you can check our Snowmass proceedings.

  • validity: -
  • significance: -
  • originality: -
  • clarity: -
  • formatting: -
  • grammar: -

Author:  Ahmed Youssef  on 2022-08-26  [id 2760]

(in reply to Report 1 by Tilman Plehn on 2022-05-27)
Category:
answer to question

We thank the referee for the careful reading of the manuscript and constructive comments. Please find below our responses to the comments and questions:

  • “in the introduction, the authors are missing the most modern generative networks based on normalizing flows. I also do not understand the point (iii) about limiting training data and coarse-grain detail being a case for a VAE;”

We thank the referee for pointing out that the references are missing. We have improved the citations. Furthermore, we have also modified the sentence that introduces the three challenges for ML based descriptions of hadronization. Hopefully, this makes it clearer what the challenges are and that we do not claim that the VAE frameworks are the only possibility. Indeed, a priori it is not clear that either GAN or NF-based architectures could not have advantages over the cSWAE architecture that we developed.

  • “also, please introduce the concept of conditional generative networks in the introduction, there are examples for conditional GANs and conditional INNs from my group, and I am sure there are more;”

Following the referee’s suggestion about the omission, we have now introduced the concept of conditional generative networks in the introduction.

  • “I appreciate the nice physics discussion in Sec.2.1, any chance you could already introduce the ML-task there? Is is just to learn Eq.(3), in essence? This might be a good point to go through ML-approaches to parton showers and compare the ways one can train a network on a simple Markov problem;”

We improved the introduction of Section 2.1 to explain that both parton shower and hadronization can be well described using Markov chain frameworks. We also added a sentence to the end of the second paragraph of section 2.1, to emphasize what the goal of the ML framework for hadronization is.

  • “Looking at Eq.(3), as a non-expert I do not really see where this formula lacks expressivity and needs to be modeled by a NN, could you please explain this a little?”

Eq. (3) does not encompass the whole complexity of the Pythia Lund model, which has O(20) parameters to describe the Lund fragmentation functions and O(200) for the complete hadronization model that includes color reconnections and multiple string interactions. A brief discussion of the drawbacks of Pythia Lund model is below Eq. (3), and we have also added two explanatory sentences at the end of Section 2.1 about what we want to achieve with the ML model of hadronization.

  • “Section 2.2 is really hard to understand. The problem might be that it mixes an introduction of the cSWAE with the physics problem and the training data. Please separate these aspects, introduce the WAE, the conditional aspects, the physics problem, and the detailed implementation and training dataset. It probably means just reshuffling paragraphs and sentences a little, but it would make things much clearer;”

Section 2.2 has now been split into two subsections (2.2 and 2.3). Section 2.2 details the architecture while section 2.3 focuses on training.

  • “in Eq.(6), please explain the mix of L1 and L2 norms, this looks interesting for a broader audience;”

The L2 norm is more sensitive to outliers compared to L1 since the difference between the incorrectly predicted target value and the correct value is squared. L2 tries to adjust the model for the outliers for the cost of the other samples. L1 norm is more robust and is not affected by outliers strongly. The combination of the two gives a more robust and stable solution. We have added a comment about this also in the draft, below eq. (9).

  • “for now, I am not sure I buy the argument with the custom distributions in latent space. The kinematical patterns I see should be easily mapped to a Gaussian. Why is my intuition wrong there? And what does this mean quantitatively? Related to this question, what am I learning from Figs.8 and 9? I am sorry, but I am missing the point there;”

Our experience was that the training performance was improved significantly when using non-gaussian latent space distributions. The custom latent space distributions allow for more flexibility, and as one can see from Table 1, it is hard to guess a priori which will be the optimal one(for instance, Pythia generated latent space distribution required, rather surprisingly, relatively large values of d_z.)

  • “in Tab.1, please explain what d_z, lambda, L mean or where to find the definitions;”

We thank the referee for pointing out the missing explanations. We have included the definitions in the corresponding caption.

  • “Figs.11 and 12 are very convincing as a bottom line, but this is also not yet an ML-hadronization model. What are the next steps after this proof of concept? Again, as a non-expert I am confused by the focus on the first emission. What happens if I train a network on all the second emissions? Please provide some more context for a broader audience;”

We have added two clarification sentences in the last paragraph of the conclusions. The focus on first emission in this manuscript is a consequence of the Markov chain nature of the process of string fragmentation. The correct description of the first emission (for any string energy, in its rest frame) is all that is required to correctly describe the full chain of string fragmentations. We train on first emission because this is available in pythia, while the agreement with full hadron spectra and the MLhad generated ones give us confidence that training on full hadron spectra will be possible. However, this still needs to be demonstrated, and in the next step, we are trying to build a training pipeline with a number of observables that would lead to an improved description of hadronization.

  • “the references are a little weird the way they are picked. Some mix of generative networks and networks used in generators. From my group, I would also have expected the original GAN paper rather than [22]. And maybe something on conditional networks. Maybe some detector simulations, like Caloflow, and definitely the generative shower models? For an overview, you can check our Snowmass proceedings.”

Following the suggestion from the referee, we have improved the citations and included the citations to the generative showers, detector simulation, and Caloflow in the introduction.

Tilman Plehn  on 2022-08-30  [id 2773]

(in reply to Ahmed Youssef on 2022-08-26 [id 2760])

Thank you for considering all my comments. I think the paper can be published, and I am very much looking forward to follow-up projects!

---

## Round 1 · Referee Report · Anonymous (Referee 2) · 2022-7-3

Report

This paper introduces a deep generative model to emulate the string hadronization model. Hadronization is currently not understood from first principles and physics-inspired phenological models are the standard. This seems like a natural place for more flexible, machine learning based solutions. The paper is thus timely and important. There are clearly many steps before we have a full ML-based hadronization model, but this work represents a key first step. I would be happy to recommend publication if the authors can address my comments and questions below.

- "Monte Carlo event generator factorizes into three distinct steps" -> what about detector simulation? Shouldn't there be four steps?

- "with significant efforts devoted to improving the precision even further" -> perhaps this should be accompanied with some references?

- "clustering model" -> isn't is usually called the "cluster" model?

- "a proof of principle that building a full-fledged ML based hadronization framework is possible." -> seems like a more precise statement would be "a first/important step towards a building a full-fledged ML based hadronization framework"? (surely a "fully fledged" framework should do more than pions, etc.)

- "In principle, both Generative Adversarial Networks (GANs) [32] and Variational AutoEncoders (VAEs)..." -> why not mention normalizing flows?

- "introduces three unique challenges" -> I don't think these are unique to hadronization. How can you produce 10^4 particles from one fragmentating parton?

- "We expect that the first version of the cSWAE architecture presented here can be upgraded to eventually be trained directly on data." -> this is a bold claim - would you please add some additional information about the steps required to achieve this ultimate goal? Some of this comes in the conclusions, but it may be useful to briefly qualify this statement or at least forward-reference. In fact, I wonder if the "conclusions" are more like "Conclusions and Outlook" since there is a lot of (useful!) information about what steps would be required to build a full-fledged model.

- "via a QCD string" -> this is jargon, would you please explain?

- "match empirical data" -> what is the word "empirical" doing here?

- How is z in Eq. 3 related to the two variables you are trying to model with the cSWAE?

- "The motivation for using cSWAE is two-fold..." -> this motivation seems to be rather generic and not specific to a cSWAE. I would have thought here you would highlight why you chose a cSWAE over other AEs and over other generative models. Would you please comment on that? (Note added: I see you discuss why SWAE instead of AE later, but maybe some of that could be discussed also here?)

- Sec. 2.2: This section is a bit hard to follow for someone not already familiar with SWAE. I know you give some details in the appendix, but I think it would be useful to already give some of them here to motivate what is being done.

- Fig. 4: What is the purpose of this figure?

- Many of the figure captions are distractingly long (e.g. Fig. 3/5). Please consider streamlining them.

- "We randomly choose one of these as the actual hadron kinematics..." -> I am confused by this - you generate 100 possibilities and then only choose one? Why not just generate one? Isn't this rather inefficient and making the problem much harder than it needs to be?

- "Pythia output on average" -> why not modify Pythia so that the MLHad can reproduce it exactly (at least in principle")? Presumably a full generative hadronization model would also be able to have some emission dependence?

- Why train for a fixed number of epochs (and not use e.g. early stopping)?

- Fig. 11: Why such big error bars? Can't these be made arbitrarily small?

- "appears to be well suited for modeling the nonperturbative process of hadronization" -> wouldn't it be more precise to say "well suited for modeling the string hadronization model"? I don't think you have demonstrated that it is well suited for "the nonperturbative process of hadronization"

- Last paragraph: aren't the public data mostly 1D binned histograms? How would you be able to "capturing the many observables in principle available in the data"?

  • validity: -
  • significance: -
  • originality: -
  • clarity: -
  • formatting: -
  • grammar: -

Author:  Ahmed Youssef  on 2022-08-26  [id 2761]

(in reply to Report 2 on 2022-07-03)

We thank the referee for the careful reading of the manuscript and constructive comments. Please find below our responses to the comments and questions:

  • “ "Monte Carlo event generator factorizes into three distinct steps" -> what about detector simulation? Shouldn't there be four steps?”

We agree with the referee that detector simulation is a critical step when working within an experimental context. However, because Monte Carlo event generators are oftentimes used for calculations without detector simulation, we do not consider detector simulation as part of the event generator itself. Here, we are discussing the steps that occur within Pythia, Sherpa, or Herwig, and not the steps that are required in something like GEANT.

-” "with significant efforts devoted to improving the precision even further" -> perhaps this should be accompanied with some references?”

We have added references to recent automated NLO work, as well as updated parton showers.

  • "clustering model" -> isn't is usually called the "cluster" model?

We thank the referee for pointing out this typographical error. We have corrected it to “cluster” model.

  • "a proof of principle that building a full-fledged ML based hadronization framework is possible." -> seems like a more precise statement would be "a first/important step towards a building a full-fledged ML based hadronization framework"? (surely a "fully fledged" framework should do more than pions, etc.)

We agree with the referee that the suggested statements are more appropriate/correct. We have changed the corresponding sentence on p.2 to “The present manuscript represents the first step toward building a full-fledged ML based hadronization framework.”

-" "In principle, both Generative Adversarial Networks (GANs) [32] and Variational AutoEncoders (VAEs)..." -> why not mention normalizing flows?"

We have changed the introduction so that we also mention the normalizing flows as a commonly used generative model. In fact, we are working right now on using normalizing flows as an alternative architecture for MLhad, with promising preliminary results.

-" "introduces three unique challenges" -> I don't think these are unique to hadronization. How can you produce 10^4 particles from one fragmentating parton?"

We agree these are not unique to hadronization and have removed “unique”. Our 10^4 claim is not for a single string, but rather for an entire event. We agree that this multiplicity is certainly rare in LHC events, but is not uncommon in central heavy ion events.

  • “ "We expect that the first version of the cSWAE architecture presented here can be upgraded to eventually be trained directly on data." -> this is a bold claim - would you please add some additional information about the steps required to achieve this ultimate goal? Some of this comes in the conclusions, but it may be useful to briefly qualify this statement or at least forward-reference. In fact, I wonder if the "conclusions" are more like "Conclusions and Outlook" since there is a lot of (useful!) information about what steps would be required to build a full-fledged model.”

We have changed “Conclusion” to “Conclusion and Outlook”, and extended the discussion of the steps that we are trying to do next.

  • ""via a QCD string" -> this is jargon, would you please explain?"

Here we mean a color flux tube between the partons. We have now introduced this in the paragraph above where we first use “QCD string”.

-" "match empirical data" -> what is the word "empirical" doing here?"

We thank the referee for pointing out the imprecise language. We have changed “empirical” to “experimental” data.

  • How is z in Eq. 3 related to the two variables you are trying to model with the cSWAE?

A clarification has been added in the text below Eq.(2).

  • “ "The motivation for using cSWAE is two-fold..." -> this motivation seems to be rather generic and not specific to a cSWAE. I would have thought here you would highlight why you chose a cSWAE over other AEs and over other generative models. Would you please comment on that? (Note added: I see you discuss why SWAE instead of AE later, but maybe some of that could be discussed also here?)”

Following the suggestion of the referee, we added a sentence toward the end of the first paragraph in Sec. 2.2, highlighting the main advantage of a SWAE.

  • “Sec. 2.2: This section is a bit hard to follow for someone not already familiar with SWAE. I know you give some details in the appendix, but I think it would be useful to already give some of them here to motivate what is being done.”

We appreciate the comment from the referee about the readability of Section 2.2. We have added extra explanatory sentences at the end of the first paragraph of Section 2.2. We have also expanded the discussion of the sliced Wasserstein distance on p. 9.

  • “Fig. 4: What is the purpose of this figure?”

Figure 4 is an illustration of how the conditional data is encoded in the latent space. Depending on the condition, it gets embedded in a different area. The purpose of the figure is to make the encoding part more clear.

  • “Many of the figure captions are distractingly long (e.g. Fig. 3/5). Please consider streamlining them.”

We have reduced the text for captions that were particularly long. In some cases, the captions do still remain long, but we believe they necessarily describe the figure.

  • “ "We randomly choose one of these as the actual hadron kinematics..." -> I am confused by this - you generate 100 possibilities and then only choose one? Why not just generate one? Isn't this rather inefficient and making the problem much harder than it needs to be?”

We generate only one point in the latent space (i.e. an ordered vector with dimensions equivalent to the latent dimension - typically ~5 - containing elements sampled from the chosen latent distribution) which is fed into the decoder and spits out an ordered 100-dimensional vector distributed according to the desired kinematic distribution. We choose one element in this 100-dimensional vector to be used as the kinematic value for the output hadron. This is done for each hadron in the hadronization process because there is a (trivial) energy dependence which changes with each fragmentation event and in principle a transverse mass dependence. The generation of the 100 kinematic possibilities is not inefficiency but a quirk of the architecture.

  • “ "Pythia output on average" -> why not modify Pythia so that the MLHad can reproduce it exactly (at least in principle")? Presumably a full generative hadronization model would also be able to have some emission dependence?”

The referee is correct that one could try to reproduce Pythia exactly by reproducing separately both the first emission (which is treated differently in Pythia) and then all the other emissions. However, the ultimate goal is not to reproduce Pythia, especially since such choices made in Pythia are not physically observable. The ultimate goal is to reproduce the physical observables, such as multiplicities, angular distributions, etc. We have thus opted in MLhad for a common treatment of the first and the subsequent emissions such that the observable quantities are reproduced.

  • " Why train for a fixed number of epochs (and not use e.g. early stopping)?"

For the datasets we are training on in this paper, a very good model can be trained in < 30 minutes so the need for more sophisticated training termination is unnecessary. This will be implemented in the future, however.

  • “Fig. 11: Why such big error bars? Can't these be made arbitrarily small?”

The referee is correct that one could spend more computer time to make the error bars imperceptibly small. However, for our purposes, the errors are already very small, much smaller than needed to show that a) there are some statistically significant differences in the fragmentation chains between Pythia and MLhad and b) that these differences are in general small.

  • “ "appears to be well suited for modeling the nonperturbative process of hadronization" -> wouldn't it be more precise to say "well suited for modeling the string hadronization model"? I don't think you have demonstrated that it is well suited for "the nonperturbative process of hadronization" “

While we take to heart the comment by the referee, we have decided, after some internal deliberation, to leave the statement as is. Strictly speaking, the statement is correct, since the string hadronization as implemented in Pythia does model well the nonperturbative process of hadronization. MLhad is constructed such that it can reproduce the Pythia model, but with additional flexibilities. It is therefore well suited to model the nonperturbative process of hadronization (as is Pythia).

  • “Last paragraph: aren't the public data mostly 1D binned histograms? How would you be able to "capturing the many observables in principle available in the data"?”

Yes, this is correct, and so we have added another sentence here clarifying the statement. Namely, we intend to use LHC open data, as well as the possibility of accessing BaBar and LEP data to construct some of these observables that are not currently available through the standard 1D histograms.

---

## Round 2 · Referee Report · Anonymous (Referee 3) · 2022-9-6

Report

Thank you for taking into account my feedback. I still not really understand why you need to generate 100 configurations and only pick one (it does seem to be an inefficiency and not merely a "quirk" of the approach), but I won't insist further.

---

## Round 2 · Author Response

We thank the referees for their careful reading of the manuscript and constructive comments. Below please find our answers to the referees, as well as the list of changes made.

---

## Round 2 · List of Changes

General changes:
We have updated the description of the figures.
Section 2.2 has been split into two subsections (2.2 and 2.3). Section 2.2 details the architecture while section 2.3 focuses on training.
We have changed “Conclusion” to “Conclusion and Outlook”

Changes in the introduction on page 2:
We have added references to recent automated NLO work and updated parton showers in the first paragraph of the introduction.
We have added references for ML-based simulations for the parton shower and detector simulation.
We reformulated “The present manuscript represents a proof of principle that building a full-fledged ML based hadronization framework is possible.” to “The present manuscript represents [the first step toward building a full-fledged ML based hadronization framework. ”
We added normalizing flows as a commonly used generative model.
We added the sentence “In addition, conditional generative models give more flexibility and control of the output” with corresponding references to introduce the concept of conditional generative models.
In the last paragraph of page 2, we removed the word “unique” for the challenges.

Changes page 3:
We added the last sentence in the first paragraph to point the reader to further steps.
We added the first paragraph in sec. 2.1 to explain that both parton shower and hadronization can be well described using Markov chain frameworks. We also added a sentence to the end of the second paragraph of section 2.1, to emphasize what the goal of the ML framework for hadronization is.
We changed “ QCD string” to “QCD color flux tubes, or strings”

Changes page 4:
We added a sentence at the beginning of page 4 highlighting the goal of our ML framework.

Changes page 5
We have added a clarification sentence above eq. 3 on how eq. 3 relates to the variables we train on.

Changes on page 6
We have added two explanatory sentences at the end of Section 2.1 about what we want to achieve with ML model of hadronization.
We have added extra explanatory sentences for the SWD at the end of the first paragraph of Section 2.2.

Changes on page 8:
We have added a comment about the norm used in the lost function below eq. (9) and expanded the discussion of the sliced Wasserstein distance

Changes on page 11:
We have added a description of the variables in table 1.

Changes on page 18
We extended the discussion about possible observables and further steps in the last paragraph in “Conclusion and Outlook”.

---

## Editorial Decision

published